# A Study on the Non-Linear Impact of Digital Technology Innovation on Carbon Emissions in the Transportation Industry

**DOI:** 10.3390/ijerph191912432

**Published:** 2022-09-29

**Authors:** Xiaoqin Chen, Shenya Mao, Siqi Lv, Zhong Fang

**Affiliations:** 1School of Economics, Fujian Normal University, Fuzhou 350007, China; 2Department of International Economics and Trade, School of Business, Wuyi University, Wuyishan 354300, China

**Keywords:** digital technology, digital innovation, transportation industry, carbon emissions

## Abstract

Transportation is an important part of social and economic development and is also a typical high-energy and high-emissions industry. Achieving low-carbon development in the transportation industry is a much-needed requirement and the only way to achieve high-quality development. Therefore, based on the relevant data of 30 provinces in China from 2010 to 2018, this research uses the static panel model, panel threshold model and spatial Durbin model to conduct an empirical study on the impact and mechanism of digital innovation on carbon emissions in the transportation industry, and draws the following conclusions. (1) Carbon emissions in the transportation industry have dynamic and continuous adjustment characteristics. (2) There is a significant inverted U-shape non-linear relationship between the level of digital innovation and carbon emissions in the industry. In regions with a low level of digital innovation, the application of digital technology increases carbon emissions in this industry, but as the level of digital innovation continues to increase its application suppresses carbon emissions, showing an effect of carbon emission reduction. (3) The impact of digital innovation on carbon emissions in the transportation industry has a spatial spillover effect, and its level in one province significantly impacts carbon emissions in other provinces’ transportation industry through the spatial spillover effect. Therefore, it is recommended to further strengthen the exchange and cooperation of digital innovation in the transportation industry between regions, improve the scale of digitalization in this industry, and accelerate its green transformation through digital innovation, thus promoting the green, low-carbon, and sustainable development of China’s economy.

## 1. Introduction

Carbon peaking and carbon neutrality are the most important national strategies in China at present. To cope with the deteriorating ecological environment, the government proposed the “double carbon” target at the 75th session of the United Nations General Assembly and included “a steady decrease in carbon emissions after reaching the peak” in China’s 2035 vision. The 14th Five-Year Plan further clarified that an action plan to reach the carbon peak by 2030 should be formulated. At the 9th meeting of the Central Finance and Economics Commission and the 29th collective study of the Central Political Bureau, General Secretary Xi Jinping further put forward higher requirements for green and low-carbon development in the current and future periods, emphasizing the need to follow the path of green, low-carbon, and high-quality development. A long-term perspective can help comprehensively reduce the carbon emissions of each industry, which forms the solemn commitment of China’s economic development and its economy’s sustainable advancement.

The transportation industry typically emits high energy and high emissions and is one sector that may restrict the achievement of the “double carbon” target [1]. With the accelerating urbanization process and the increasing inter-regional movement of materials and people, the total carbon emissions of the transportation sector and its growth rate are both showing a rapid expansion trend [2]. 2020′s “China’s Carbon Peak by 2030 Study Report” pointed out that China’s transportation sector carbon emissions are currently 860 million tons, accounting for 9% of energy carbon emissions, with an average annual growth rate of 5%. It is expected by 2030 that its carbon emissions from the transportation sector will continue to grow by more than 50%, and the carbon peak of the transportation sector will be achieved later than that of the industrial and construction sectors and only after the overall carbon peak of the country [3,4]. Therefore, to achieve the “double carbon” target as soon as possible, the green and low-carbon development of the transportation sector is essential.

In the “digital” era, all industries are facing new opportunities for low-carbon development. Cloud computing, big data, artificial intelligence, the Internet of Things, and other digital technologies enable the formation of a low-carbon transformation industrial model that can promote the digitalization of key industries and fields along with green integration development [5]. China has emphasized the leading role of technological innovation in the transportation industry, proposing to accelerate the construction of a modern transportation system and to improve the construction of green and low-carbon transportation infrastructure based on convenience, high speed, light weight, and high technology [6,7]. The European Commission has also proposed to increase innovation in the use of new technologies to promote carbon emission reduction in the transport sector.

Digital innovation helping reduce carbon emissions in the transportation industry has become the current trend. The Chinese government will continue to promote the integration of digital technology and transportation systems in the future and further strengthen the exchange and cooperation of digital innovation in the transportation industry between regions [8,9]. In the existing literature, the impact of digital innovation on the carbon emissions of the transportation industry is still uncertain, so this paper is committed to research on the impact of carbon emissions of the transportation industry, to analyze what kind of impact digital innovation level has on the carbon emissions of the transportation industry, and how the specific mechanism is. As an important energy consumption and carbon dioxide emission department, it is of great significance for China to clarify the impact mechanism of digital technology innovation on carbon emissions of the transportation industry to achieve the national carbon peak goal by 2030. In addition, some of the literature has proposed the spatial characteristics of carbon emissions, but ignored the spatial correlation of the impact of information and communication technology and the Internet on carbon emissions in the transportation industry, lacking the research on spatial spillover effects. Therefore, this paper further studies the spatio-temporal evolution characteristics of digital innovation level on carbon emissions in the transportation industry in combination with the spatial autocorrelation and spatial spillover effects of China’s carbon emissions. In summary, the research herein analyzes the impact mechanism of digital innovation on carbon emissions in the transportation industry and further explores the spatial effects of carbon emissions in the industry, so that national government departments can better set policy goals, fully exploit the emission reduction potential of this industry, enjoy the dividends of digital economy development, and contribute to the construction of a strong ecological civilization and comprehensive sustainable development for China.

The remainder of this study is organized as follows. Section 2 describes the research hypothesis. In Section 3, we explain the model specification and variable selection. The empirical results and discussion are represented in Section 4. Section 5 concludes the conclusions and shows us the policy implications.

## 2. Research Hypothesis

### 2.1. The Impact of Digital Technology Innovation on Carbon Emissions in the Transportation Industry and Its Non-Linear Characteristic

As a major carbon-emitting country, China has proposed to reach the carbon peak by 2030 and achieve carbon neutrality as early as possible. With the rapid development of the digital economy, many scholars have started to incorporate research and development (R&D) investment and technological innovation into the study of carbon emissions, confirming that technological progress and innovation are the key links to achieve their reduction [10,11]. The theory of endogenous technological progress states there is a bias in technological progress [12]. Some scholars classify technologies into different types, such as clean and polluting technologies [13], environmental technologies, energy use technologies, capital embodied technologies, and broad technologies [14]. Hao et al. (2021) concluded through empirical analysis that environmental technology advancement and capital-embodied technology advancement can reduce carbon emissions, while broad technology advancement and energy use technology advancement further increase carbon emissions. Yin et al. (2022) noted that anthropogenic production technology effects can reduce carbon emissions in the industrial sector, while structural production technology promotes carbon emissions in medium- and low-emission industries [15]. Therefore, digital technology innovation is inextricably linked to carbon emission reduction in the transportation sector, which has high energy consumption characteristics.

Regarding the relationship between digital technology innovation and carbon emissions, Liang et al. (2017) argued that technological progress brought by R&D investment is the key to achieving carbon emission reduction, and this relationship is non-linear and has different characteristics in the central and eastern regions of China [16]. Digital innovation can improve the development of the whole transportation industry by promoting economic development, thus increasing the use of various modes of turnover such as freight, passenger, and shipping in a province and its neighboring provinces. Thus, carbon emissions from transportation increased significantly in the pre-development period, and then with the development of the digital economy, digital innovation optimized the energy by reducing the consumption of traditional fossil energy in the transportation industry [17,18]. With the development of the digital economy, digital innovation is optimizing the energy use structure by reducing the consumption of traditional fossil energy in the transportation industry and gradually replacing the use of traditional energy from new energy sources, which in turn lower total carbon emissions in the transportation industry. Based on this, this paper proposes Hypothesis 1.

**Hypothesis** **1 (H1).**
*Digital technology innovation impacts carbon emissions in the transportation industry and has a non-linear characteristic.*


### 2.2. Spatial Spillover Effects of Carbon Emissions from the Transportation Sector

The digital economy allows the sharing of information and development dividends through technological innovation and enhances the linkage of economic activities between provinces and regions. Jin et al. (2022) concluded that once China’s digital economy linkage network took its initial shape, the agglomeration effect and spill-over effect of provinces gradually increased, and the mobility of provincial digital resource elements rose significantly [19]. Bi et al. (2019) studied the impact of digital technology on environmental quality in China using a spatial model and found that technological progress reduces the emission of pollutants and there is a spatial spillover effect [20]. Xu et al. (2022) constructed a dynamic spatial Durbin model based on the spatial autocorrelation test to study the spatial effect of strong carbon emissions in China, noting that technology input produces a negative spatial spill-over on carbon emission intensity [21]. Analyzing China’s transportation carbon emissions data from 1996–2014, Zhang et al. (2019) found a spatial spill-over effect of transportation carbon emissions in some provinces and a strong spatial correlation within provinces [22]. It is logical that the impact of digital technology on carbon emissions in the transportation industry should also have a spatial spill-over effect. Therefore, this paper proposes the Hypothesis 2.

**Hypothesis** **2 (H2).**
*Digital technology innovation has a spatial spill-over effect on carbon emissions in the transportation industry.*


### 2.3. Specific Mechanisms of the Role of Digital Technology Innovation on Carbon Emissions in the Transportation Industry

According to the technological innovation effect of the digital economy, there are two main paths whereby the digital economy affects carbon emissions in the transportation industry through technological innovation. On the one hand, the digital interconnection of various industries, the significant increase in passenger and cargo turnover, and the emergence of various modes of transportation have promoted the development of the transportation industry and further increased its total carbon emissions [23,24]. The development of this industry has pushed up carbon emissions, and there is a significant spatial positive relationship between its carbon emissions and motor vehicle ownership, GDP, freight turnover, and passenger turnover [25]. On the other hand, digital technology can optimize the energy use structure of the transportation industry. Zhang et al. (2022) concluded that the digital economy improves the energy structure by reducing the use of traditional fossil energy, and this improvement is an important mechanism for the reduction of carbon emission intensity caused by the digital economy’s development [26].

In terms of energy consumption structure, gasoline, kerosene, diesel, and fuel oil account for the majority of such consumption in the transportation industry, and the energy structure closely relates to carbon emissions. As digital interconnection promotes the connection between innovation subjects and innovation sharing, it also generates digital technology empowerment. The in-depth application of digital technology helps change people’s traditional concept of fossil energy consumption, generate new ways of energy consumption, facilitate the promotion of clean energy and the use of green technologies to areas with serious environmental pollution, and promotes the transformation of energy consumption in the transportation industry to further spur the use of new energy sources and digital technology applications in the field of new energy industry that can maximize the digital energy innovation mechanism [27,28]. The application of digital technology in the new energy industry can maximize the driving effect of digital energy innovation mechanisms on energy restructuring [29,30]. The application of new energy to all aspects of transportation promotes the green transformation of transportation energy and helps achieve its carbon reduction effect. The Hypothesis 3 is now presented.

**Hypothesis** **3 (H3).**
*The level of transportation development has a positive moderating effect in digital technology innovation’s impact on carbon emissions in the transportation industry. The structure of transportation energy consumption has a negative moderating effect in digital technology innovation’s impact on carbon emissions in the transportation industry.*


## 3. Model Specification, Variables, and Data Description

### 3.1. Model Specification

#### 3.1.1. Modeling of the Static Panel

Since total economic growth per capita, energy consumption intensity, urbanization level, openness to the outside world, regional consumption capacity, transportation structure, and industrial structure have an impact on carbon emissions in the transportation industry, these factors are used as control variables to explore digital innovation’s influence on energy carbon emissions in the transportation industry in a targeted manner. To eliminate heteroskedasticity, the explanatory variables and control variables are logarithmically treated. Therefore, the paper sets up the following static panel data model.
(1)$$\ln c_{i,t}=\alpha_{i}+\beta_{1}dig_{i,t}+\beta_{2}controls_{i,t}+\varepsilon_{i,t}$$

Here, *nc*_*i*,*t*_ denotes the natural logarithm of total carbon emissions in the transportation sector in year *t* for province *i*; *dig*_*i*,*t*_ denotes the digital innovation level in province *i* in year *t*; *controls*_*i*,*t*_ denotes all control variables in province *I* in year *t*, including total economic growth per capita, urbanization level, openness to the outside world, regional consumption capacity, and industrial structure; *α_i_* denotes unobservable individual effects; and *ε*_*i*,*t*_ is a random error term.

#### 3.1.2. Modeling of the Panel Threshold

The previous hypothesis states that carbon emissions from the transportation sector will change with an increase in the level of digital innovation. To further investigate this non-linear relationship, the study sets up the following panel threshold model.
(2)$$\ln c_{i,t}=\alpha_{i}+\lambda_{1}dig_{i,t}\times I(Adj_{i,t}\leq \theta )+\lambda_{2}dig_{i,t}\times I(Adj>\theta )\beta_{2}c\times \beta_{2}controls_{i,t}+\varepsilon_{i,t}$$

Here, *Adj*_*i*,*t*_ is the threshold variable for the digital technology innovation; and *I*(•) is the representative indicative function with a value of 1 when the condition in parentheses is satisfied and otherwise 0. When the coefficients *λ*_1_ and *λ*_2_ are not equal, it means that there is a threshold effect, and the rest of the variables are the same as in Equation (1).

#### 3.1.3. Modeling of the Spatial Dependence

As neighboring provinces tend to have similar economic structures and life characteristics, the carbon emissions and influencing factors in the transportation sector between them also correlate. To further discuss the spatial spillover effect of digital innovation level on carbon emissions in the transportation industry, this paper introduces the spatial interaction terms of explanatory variables,, and control variables based on the benchmark model as follows.
(3)$$\ln c_{i,t}=\alpha_{i}+\rho Whc_{i,t}+\varphi_{1}Wdig_{i,t}+\beta_{1}dig_{i,t}+\varphi_{2}Wdig2_{i,t}+\beta_{2}dig2_{i,t}+\varphi_{3}Wcontrols_{i,t}+\beta_{3}controls_{i,t}+\varepsilon_{i,t}$$

Here, *ρ* is the spatial autoregressive coefficient—i.e., the carbon emission effect of a province caused by other neighboring provinces; *W* is the spatial weight matrix, and the three methods of economic distance, geographic distance matrix, and adjacency matrix are used herein for regression to improve the robustness of the empirical results; and *φ*_1_, *φ*_2_ and *φ*_3_ are the coefficients of the core explanatory variables as well as the spatial interaction terms of the control variables. Equation (3) contains the spatial interaction terms of the explanatory and explanatory variables and is known as the spatial Durbin model (SDM).

The economic distance matrix is constructed based on the GDP per capita of China from 2010–2018. The geographical distance matrix is constructed based on the inverse distance matrix with the Euclidean distance calculated from the latitude and longitude of the provincial capital cities. The adjacency matrix at *W* is denoted as follows.
(4)$$W=\left\{ \begin{array}{l} 1,\;\mathrm{If}\;\mathrm{the}\;\mathrm{two}\;\mathrm{provinces}\;\mathrm{are}\;\mathrm{adjacent}\;\mathrm{to}\;\mathrm{each}\;\mathrm{other};\\ 0,\;Qthers. \end{array}\right.$$

### 3.2. Variables and Data Description

#### 3.2.1. Explained Variables

For carbon emissions from transportation (CO_2_), this paper adopts the top-down method of energy end-use consumption and estimates the carbon emissions from transportation in 30 provinces and cities (except Tibet, Hong Kong, Macao, and Taiwan) during 2010–2018 based on the amount of energy end-use consumption and CO_2_ emission coefficients in transportation. Currently, energy consumption indicators in transportation are often reported together with storage and postal industries. Considering the low proportion of storage and postal industries, this paper uses the energy consumption data of transportation, storage, and postal industries from the China Energy Statistical Yearbook for each year. This paper divides final energy consumption into nine categories, including eight types of fossil energy consumption and electricity consumption in the secondary energy. Various energy sources are first converted into standard coal in the calculation of carbon emissions.

Carbon emissions are estimated according to the method provided by the 2006 IPCC-designated Guidelines for National Greenhouse Gas Emissions Inventories with the following formula.
(5)$${\mathrm{CO}}_{2}=\sum_{i=1}^{9}{\mathrm{CO}}_{2}=\sum_{i=1}^{9}E_{i}\times SCC_{i}\times CEF_{i}$$

Here, CO_2_ denotes estimated carbon emissions; *i* denotes each energy source; *E_i_* denotes energy consumption; *SCC_i_* is the discounted standard coal factor for each energy source; and *CEF_i_* is the carbon emission factor provided by IPCC 2006. The details are in Table 1.

#### 3.2.2. Core Explanatory Variables

In terms of digital innovation level (dig), with the support and guidance of many national policies and the comprehensive effect of new-generation information technology, China’s Internet industry has developed rapidly and paid more attention to R&D investment. At the same time, R&D innovation has gradually shown its leading role in the digital industry, with this new industry and new mode of the Internet developing rapidly, expanding in scale and in innovation and pushing digital innovation capability as the key to the digital economy’s advancement. This paper draws on one of the core dimensions of the digital economy based on “digital technology innovation level” [31,32] and selects the volume of full-time R&D personnel, R&D funding, number of R&D projects, and number of digital economy-related patent applications (from the patent search website of the State Intellectual Property Office) to measure the level of digital innovation. That is, the level of digital innovation is a comprehensive indicator of all variables shown in Table 2. The specific evaluation system appears in Table 2.

#### 3.2.3. Control Variables

Carbon emissions from the transportation industry are bound to be influenced by various other factors. With economic development, per capita car ownership will increase and people’s travel demand will rise sharply, and so economic development will raise national transportation carbon emissions [33,34]. As China’s openness increases and urbanization advances, economic relations around the country become increasingly close, transportation activities become more frequent, and carbon emissions from the transportation industry appear to change spontaneously and non-spontaneously. In general, the greater the proportion is of the added value of the transportation industry to the added value of the total industry, the greater the effect of promoting emission reduction in the transportation industry. To more accurately analyze the impact of digital innovation on the total carbon emissions of the transportation industry, the following control variables are included in this paper.

Economic development level (pgdp): In this paper, GDP per capita is selected as the economic development level indicator, and both it and GDP are obtained from the statistical yearbooks of each province.

Openness to the outside world (open): This paper selects the ratio of the total imports and exports of each domestic province to the total value of the country’s economic growth, where the total import and export is converted to RMB using the annual exchange rate. Total imports and exports are obtained from the statistical yearbook of each province, and the annual average exchange rate of China is obtained from the China Statistical Yearbook.

Urbanization level (urban): This denotes the share of urban population in the total regional population in each province in each year.

Regional consumption capacity (consume): It is the proportion of total retail sales of social consumer goods to China’s GDP. Total retail sales of social consumer goods are from the statistical yearbook of each province.

Industry structure (transd): It is expressed by the value added of the transportation industry over the total value of economic growth. A higher value means a greater proportion of the transportation industry in the region.

In view of the validity and reliability of the data, the paper chooses panel data of 30 provinces and cities of China (except Tibet, Hong Kong, Macao, and Taiwan) during the nine-year period from 2010 to 2018. The descriptive statistics of each variable are in Table 3.

## 4. Results and Discussion

### 4.1. Panel Model Regression Analysis

The results of the baseline regression appear in Table 4. Column (1) reports the coefficient relationship between adding only the level of digital innovation and carbon emissions in the transportation industry, and the estimated coefficient of the core explanatory variable dig is significantly positive. Columns (2), (3), and (4) (i.e., exploring the relationship between carbon emissions in the transportation industry and digital innovation) include control variables and gradually control for time and spatial effects. For the static panel, the paper uses the Hausman test to test whether a fixed effect or random effect is used. The test result *p*-value is 0, which rejects the original hypothesis, and so the fixed effect model should be used. The results of each regression show that digital innovation significantly relates to total carbon emissions in the transportation sector, and the control variables also pass the significance test. This demonstrates that carbon emissions in the transportation sector are influenced by digital innovation.

To further test hypothesis H1, in which there is more than a simple linear relationship between digital technology innovation and carbon emissions in the transportation industry, this paper uses a panel threshold model. Before conducting the threshold effect test, it is necessary to determine whether the threshold effect exists and the number of thresholds that exist. Therefore, this study conducts the threshold existence test based on Hansen’s method, and the results show the digital technology innovation index as the threshold variable passes the single threshold test at the 95% level. Therefore, this paper sets the unit threshold for regression and presents the regression results in Table 5. The regression results show that the level of digital technology innovation development has a non-linear relationship on the impact of carbon emissions in the transportation industry, and the single threshold value with digital technology innovation as the threshold variable is 0.582, which is significant at the 5% level. When the level of digital technology innovation is below the threshold value, 0.582, an increase in the level of digital technology innovation will significantly raise the total carbon emissions of the transportation industry. With the continuous development of digital technology, when the value exceeds the threshold value, the coefficient of the level of digital technology innovation is −0.292, which indicates that each unit increase in the level of digital technology innovation at this time will reduce carbon emissions of the transportation industry by 0.292%. This is because the use of the Internet in the digital era has made people’s lives smarter, and the application of new technologies has optimized the transportation system and improved energy use efficiency. This regression result further confirms hypothesis H1.

### 4.2. Spatial Effect Analysis

In order to specifically analyze the spatial spillover effect of carbon emissions from China’s transportation industry, this paper conducts a spatial effect analysis. Before doing the spatial econometric analysis, the spatial autocorrelation test between the level of digital innovation and carbon emissions in the transportation industry is first executed. This study uses Moran’s I index to calculate the spatial effects from 2010 to 2018 under the economic distance matrix. The results are in Table 6 and Table 7. Table 6 presents that the level of digital innovation has a strong spatial dependence in all years. Table 7 shows a significant spatial correlation between the carbon emissions of the transportation industry in China for all years except 2010, which indicates the existence of spatial clustering between the two. Therefore, it is necessary to consider spatial correlation using spatial econometric models.

This paper applies the LM test, SDM fixed effects model, and Hausman test to determine which spatial econometric model to use, indicating that the two-way fixed effects SDM model is the optimal choice. Table 8 gives the estimation results of the SDM fixed effects estimation model for different distance matrices.

The specific analysis is as follows (taking the economic distance matrix as an example). The spatial lag coefficient of the digital innovation level is 1.214 with a significance level of 1%, and the coefficient of its squared term is −0.209, which is also significant at the 1% level. This indicates that digital innovation in a province not only affects local transportation carbon emissions, but also affects transportation carbon emissions in other provinces with similar levels of economic development to its own through the spatial lag effect. The coefficient of the indirect effect in (1) shows that the transportation industry of a province is also driven by the economic development and digital innovation of other provinces, which in turn affect the total carbon emissions of the transportation industry. The positive coefficient of digital innovation level and the negative coefficient of quadratic term coefficient indicate that when the digital innovation level of a province exceeds a certain threshold value, the carbon emissions of the transportation industry in that province will not only fall, but so will the carbon emissions of other neighboring provinces or areas with a similar economic development level.

### 4.3. Endogenous Problems

The endogeneity problem arises for two main reasons: one being that the independent and dependent variables causally relate to each other, and the other being the omission of important variables. Endogeneity can make the estimated coefficients biased. In order to alleviate the endogeneity problem, this study further employs the two-stage least squares method with instrumental variables and the generalized moment estimation method to solve it.

First, this paper adopts the method of Huang et al. (2019) and selects the number of post offices in 1984 in each province and city as an instrumental variable for the level of digital innovation [35]. The reason for choosing this instrumental variable is that the distribution of post offices represents the popularity of ICT (Information and Communications Technology) in the region, and post offices can influence the level and development rate of Internet technology application, which satisfies the relevance of the instrumental variable. In addition, since the instrumental variable is selected as historical data, traditional communication tools such as landline post offices have minimal impact on a region via the development of digital technology, which satisfies exclusivity. Since the original data for 1984 are cross-sectional and do not satisfy the analysis of panel data, drawing on Nunn & Qian’s method, this paper constructs the interaction term between the number of landline telephones per 10,000 people in each province in 1984 and the number of national Internet users in the previous year, respectively, as an instrumental variable for the level of digital innovation in that year [36]. Columns (1) and (2) of Table 9 report the results of the two-stage least squares regression of the instrumental variables, where the Kleibergen–Paap rk LM statistic has a *p*-value of 0, the original hypothesis of “unidentifiable instrumental variables” is rejected, and the Kleibergen–Paap rk Wald F statistic results indicate no weak instrumental variables. Column (1) of Table 9, the first-stage regression results, shows that the instrumental variables and the core explanatory variables pass the significance test at the 5% level, and the second-stage results indicate that the introduction of instrumental variables does not affect the main findings of the model.

Second, the study sets up dynamic panel models and estimates the results using generalized difference GMM and two-step systematic GMM (see columns (3) and (4) of Table 9). The Arellano–Bond statistic is chosen for the empirical analysis process to test the autocorrelation of the model. As seen from the Arellano–Bond statistic AR (2) reported in the empirical results, its *p*-value is greater than 0.1, which rejects the hypothesis of the existence of autocorrelation and indicates that the GMM estimates are consistent. The Hansen statistic is chosen for the GMM model with instrumental variables selected for the over-identification test. Here, the *p*-values corresponding to the Hansen J statistic in Table 5 are all above 0.1, indicating that the selected instrumental variables are reasonable and there is no over-identification. From the specific regression results in Table 5, we also know that the coefficients, sign direction, and significance of the variables in the generalized method of moments estimation do not differ significantly from the baseline regression, further verifying that carbon emissions from the transportation sector in the previous period positively correlate with total carbon emissions from the transportation sector in the current period at the 1% significance level, proving that carbon emissions from China’s transportation sector are path-dependent, and that China should take advantage of this characteristic of carbon emissions to adopt more efficient emission reduction programs and measures.

### 4.4. Robustness Test

Firstly, to test the robustness of the panel regression of sample data, this paper first replaces the total carbon emissions of the transportation sector (i.e., the ratio of total carbon emissions of the transportation sector to GDP) with the total carbon emissions of the transportation sector as the explanatory variable. Transportation energy intensity is one of the main influencing factors of transportation carbon emissions, and a decrease in energy consumption intensity will help total carbon emissions also decrease. With the improvement of residents’ consumption ability, people are gradually improving their physical and psychological needs, and the choice of transportation means of travel appears hierarchical. Thus, the regression results prove the robustness of the regression. Second, the measurement of digital innovation level is somewhat subjective. Hence, in order to prove the scientific credibility of the regression results, the Internet penetration rate is used instead of digital innovation level as the core explanatory variable. The Internet penetration rate is measured by the ratio of the number of Internet users to the regional resident population. Finally, because of the different resource endowments and development characteristics of each province and region, there is heterogeneity in their transportation conditions and digital innovation levels. Therefore, the impact of digital innovation on carbon emissions in the transportation industry may also be heterogeneous at the provincial level. This paper explores this angle in greater depth.

The specific regression results are shown in the following Table 10. Table 10 (1) lists the regression results of the fixed effects model after replacing the explanatory variables, (2) presents the regression results after replacing the explanatory variables, and (3) offers the robustness test of grouped regressions by dividing China into east, central, and west regions. Specifically, the correlation between digital innovation and carbon emissions in the transportation industry is greater in the central region, because the economic development there is fast in general, and digital technology and digital innovation in the central region is surging. The central region should further stimulate faster and better digital development after breaking through the threshold of digital technology innovation and enjoy the carbon emission reduction dividend brought by digital technology innovation, laying a good foundation for the green development of the transportation industry there. Development in the western region is relatively backward, and development there has enhanced the local industry to a certain extent, but digital connectivity still needs to be increased and to be popularized and better integrated with the transportation industry. Overall, the results show that the estimated coefficients of all variables have the same direction and correlate at different significance levels. The full-sample benchmark regression passes the stability test.

Secondly, the study tests the spatial effect for robustness by using a spatial autoregressive (SAR) model; i.e., only the spatial correlation of the explanatory variables is considered, and the coefficient of the spatial lag term is 0. The results appear in Table 11 and confirm that digital innovation in a provincial area not only has an impact on its own transportation carbon emissions, but also promotes or inhibits the other provinces. A specific impact change has a turning point that shows an inverted U-shape non-linear relationship, and the spatial spillover effect also passes the robustness test.

### 4.5. Mechanism Analysis

Based on the previous research hypotheses, this study uses the total converted transport turnover by year as an indicator of the level of transport development. The share of fossil energy consumption in the end consumption of the transport sector within total energy consumption of the transport sector is a proxy variable for the energy consumption structure of the transport sector. The SDEM model helps evaluate the moderating effect of transportation development level and transportation energy consumption structure for the impact of digital innovation on carbon emissions in the transportation industry.

The results are shown in Table 12. Column 2 presents that the interaction term between the squared term of digital innovation and the level of transportation development has a significantly positive effect on carbon emissions, which indicates that as the level of transportation development improves, the direct inhibitory effect of digital innovation on carbon emissions in the transportation sector diminishes; i.e., the higher the level of transportation development, the slower the process of digital innovation on carbon emissions in the transportation sector from promotion to suppression. Furthermore, the spatial lag coefficient of the interaction term is 0.0045 with a significance level of 10%, meaning that the improvement in the level of transportation development of a province not only suppresses carbon emission reduction of digital innovation on the transportation industry, but also that of its neighboring provinces.

The results in column 3 of Table 12 illustrate that the interaction term between the squared term of digital innovation and the structure of transportation energy consumption has a significantly negative effect on total transportation carbon emissions, and that the change in people’s perceptions about the use of traditional fossil energy sources to new energy sources is crucial to the carbon emissions of the transportation sector. As seen from the results, the spatial lag coefficient of the interaction term is −0.0092 with a significance level of 1%, denoting that the transportation energy structure not only strengthens the negative direct effect of digital innovation on carbon emissions in the transportation sector, but also its negative indirect effect. This further validates the research hypothesis. Moreover, both specific transmission mechanisms, the level of transportation development, and the structure of transportation energy consumption, hold true for influencing carbon emissions in the transportation sector.

## 5. Conclusions and Policy Recommendations

The 14th Five-Year Plan is a critical period to achieve a shift in the growth rate of carbon emissions. China’s industries need to seize the opportunity of digital development and share the dividends of Internet development in order to achieve the double carbon target as scheduled. The transportation sector is a major source of global carbon emissions and thus a key area for achieving the double carbon target of energy savings and emission reduction in the future [37]. Considering the possible non-linear, dynamic, and spatial effects of digital innovation on carbon emissions in the transportation sector, this study employs dynamic panel models and spatial econometric models based on China’s provincial panel data from 2010–2018 to explore the relationship between them. The main findings are as follows.

First, carbon emissions in the transportation industry are a dynamic and continuous adjustment process and are path-dependent. There is also a significantly positive relationship between carbon emissions in the transportation industry in the previous period and carbon emissions in the current period.

Second, there is a significant inverted U-shape non-linear relationship between digital innovation and carbon emissions in the transportation industry. In regions with low levels of digital innovation, digital technology applications increase the intensity of carbon emissions in the transportation industry. However, as the level of digital innovation continues to increase, digital technology applications turn to suppress carbon emissions, showing a carbon emission reduction effect.

Third, digital innovation has a significant spatial spillover effect on transportation carbon emissions. Digital innovation in a province not only affects its own transportation carbon emissions, but also affects the transportation carbon emissions of its neighboring or economically related provinces. This effect also exhibits a non-linear characteristic.

Fourth, the impact of digital innovation level on transportation carbon emissions is inhibited by the level of transportation development. The impact is further enhanced by the structure of transportation energy consumption. It also shows spatial characteristics.

The above findings lead to the following policy recommendations.

First, China must continue to take positive action to reduce energy consumption and emissions in the transportation sector. Studies have shown that carbon emissions from the transportation sector have a dynamic continuum. In order for the transportation sector as a whole to achieve carbon emissions reduction, its energy efficiency and emission reduction policies must be continuously updated and optimized [38]. Government departments should strengthen investment in transportation infrastructure and continue to increase energy funds and subsidies to alleviate the cost pressure in upstream raw materials. Instead of blindly taking a broad-brush approach to traditional primary energy sources such as coal and oil, China should gradually reduce the use of primary energy in transportation and continue to increase the proportion of new energy consumption [39]. In addition, the carbon emission assessment of the transportation sector should be strengthened, and the concept of green travel should be promoted.

Second, relevant authorities should expand the role of digital innovation in driving carbon emissions reduction in the transportation industry. The development of China’s transportation industry should focus on digital technology, by increasing investment in science and technology innovation in this industry, continuously expanding the investment and financing channels within the digital industry, guiding the transportation industry to strengthen the importance of its R&D sector, and giving priority protection to the R&D sector in terms of funding. China could also establish good channels for cooperation between industry, academia, and research in the digital industry, introduce high-tech talents or scientific researchers from universities to join innovative activities in the transportation industry, and give certain patent fund support to relevant technical personnel to fully mobilize the digital talents’ R&D enthusiasm and initiative. Doing so could help China as a whole improve the level of digital innovation.

Third, the exchange and cooperation of digital innovation in the transportation industry between regions should be enhanced to improve the scale of digitalization of the transportation industry and promote the green, intelligent, coordinated, and sustainable development of regional transportation. Research results show that the priority development of digital technology and digital innovation will lay a good foundation for the green development of its transportation industry. The western region is relatively backward in development and carbon emission reduction in its transportation industry and has not been able to enjoy the digital innovation dividend. Thus, the central region should give full play to its own resources and economic advantages to drive the backward regions to cross the inflection point of digital innovation as fast as possible. Government departments should focus on exchange and cooperation among regions to narrow the technological differences between them. Lastly, China should pay attention to the rational deployment of resources, promote open cooperation among regions, and build a strong interconnection among industry, talent, technology, information, and other resources to achieve complementary advantages.

## Figures and Tables

**Table 1 ijerph-19-12432-t001:** Carbon emission factors and standard coal conversion factors of nine energy sources.

Energy	Carbon Emission Factor (Tons of Carbon/Tons of Standard Coal) CEF	Discount Factor for Standard Coal (kg Standard Coal/kg) SCC
Coal	0.7476	0.7143
Coke	0.1128	0.9714
Crude Oil	0.5854	1.4286
Gasoline	0.5532	1.4714
Kerosene	0.3416	1.4714
Diesel	0.5913	1.4571
Fuel Oil	0.6176	1.4286
Natural Gas	0.4479	1.3300
Power	2.2132	0.1229

**Note:** The converted quasi-coal coefficients are from the 2013 China Energy Statistics Yearbook, and the carbon emission coefficients for each energy source are from the 2006 IPCC.

**Table 2 ijerph-19-12432-t002:** Evaluation system of China’s digital innovation level.

Digital technology innovation level	Full-time volume of R&D staff R&D funding Number of R&D projects Number of digital economy-related patent applications

**Table 3 ijerph-19-12432-t003:** Results of descriptive statistics for each variable.

Variable Category	Variable Name	Variable Symbol	Variable Definition	Variable Unit	Total Number of Variables	Average Value	Standard Deviation	Minimum Value	Maximum Value
Explained variables	Transportation energy carbon emissions	CO_2_	Measured by the method provided by IPCC in 2006; the formula is (4)	million tons	270	1518.202	447.704	83.053	5079.747
Core explanatory variables	Digital Innovation level	dig	Comprehensive evaluation system according to Table 2	-	270	0.480	0.220	0.120	1.430
Control variables	Economic growth level	pgdp	Expressed as gross economic value added per capita	10,000 Yuan/person	270	5.070	2.470	1.310	14.02
	Urbanization level	urban	Total urban population/resident population	-	270	0.570	0.130	0.340	0.900
	Open to the public	open	Total imports and exports/GDP	-	270	0.250	0.320	0.000	1.550
	Regional consumption power	consume	Total retail sales of social consumer goods/GDP	-	270	0.380	0.0700	0.230	0.600
	Industry Structure	transd	Value added of transportation industry/GDP	-	270	0.050	0.010	0.020	0.100

**Table 4 ijerph-19-12432-t004:** Panel model regression results.

Variable	(1)	(2)	(3)	(4)
dig	1.9204 *** (0.2337)	1.8875 *** (0.6715)	1.8553 *** (0.5071)	2.1443 *** (0.7324)
Constant	5.1941 *** (0.0713)	5.4022 *** (0.1110)	5.9019 *** (0.4946)	5.2232 *** (0.4990)
Controls	Uncontrolled	Control	Control	Control
Time effect	No	No	No	Yes
Spatial effects	No	No	Yes	Yes
R-squared	0.3532	0.4096	0.5237	0.5021
Obs.	270	270	270	270

**Notes:** Robust standard errors are within ( ). ***, **, and * denote significance levels at 1%, 5%, and 10%, respectively.

**Table 5 ijerph-19-12432-t005:** Threshold model regression results.

Variable	dig
Threshold value	0.582
Dig × I (Th ≤ q_1_)	2.434 *** (0.173)
Dig × I (Th > q_1_)	−0.292 *** (0.0520)
Controls	Control
Constant	6.495 *** (0.144)
Obs.	270
R-squared	0.639

**Notes:** Robust standard errors are within ( ). ***, **, and * denote significance levels at 1%, 5%, and 10%, respectively.

**Table 6 ijerph-19-12432-t006:** Characteristics of the spread of digital innovation level emissions for 2010–2018.

Year	Moran’s I	Z-Value
2010	0.258 ***	2.905
2011	0.283 ***	3.142
2012	0.271 ***	3.029
2013	0.258 ***	2.902
2014	0.276 ***	3.075
2015	0.269 ***	2.996
2016	0.301 ***	3.333
2017	0.145 **	1.781
2018	0.115 *	1.483

**Notes:** ***, **, and * denote significance levels at 1%, 5%, and 10%, respectively.

**Table 7 ijerph-19-12432-t007:** Carbon emissions spread of characteristics in the transportation sector for 2010–2018.

Year	Moran’s I	Z-Value
2010	0.178	2.191
2011	0.359 ***	4.716
2012	0.349 ***	4.565
2013	0.344 ***	4.501
2014	0.359 ***	4.686
2015	0.356 ***	4.655
2016	0.337 ***	4.398
2017	0.291 **	3.333
2018	0.362 ***	4.759

**Notes:** ***, **, and * denote significance levels at 1%, 5%, and 10%, respectively.

**Table 8 ijerph-19-12432-t008:** Regression results of the spatial model of digital innovation level affecting carbon emissions in the transportation industry.

Model Setting		SDM		
Spatial Matrix Type		Economic Distance	Geographical Distance	Adjacency Matrix
Variable		(1)	(2)	(3)
rho		−0.231 ** (0.0903)	−1.020 *** (0.238)	−0.150 ** (0.0638)
dig		0.274 * (0.161)	0.337 ** (0.139)	0.370 ** (0.154)
dig2		−0.0339 (0.0266)	−0.0422 * (0.0230)	−0.0544 ** (0.0254)
W*dig		1.214 *** (0.523)	−0.943 (0.860)	0.597 * (0.326)
W*dig2		−0.209 *** (0.0720)	0.163 (0.151)	−0.0966 * (0.0551)
Controls		Yes	Yes	Yes
Direct effect	dig	0.224 * (0.175)	0.385 ** (0.150)	0.358 ** (0.160)
	dig2	−0.0276 * (0.0289)	−0.0502 ** (0.0249)	−0.0527 ** (0.0263)
Indirect effects	dig	1.157 ** (0.477)	0.660 * (0.467)	0.498 * (0.303)
	dig2	−0.157 ** (0.0674)	−0.105 * (0.0819)	−0.0819 (0.0509)
Total effect	dig	1.381 *** (0.417)	1.045 ** (0.443)	0.855 *** (0.310)
	dig2	−0.185 *** (0.0578)	−0.155 ** (0.0779)	−0.135 ** (0.0527)
LogL		231.0028	258.2256	217.8049
R-squared		0.164	0.094	0.071

**Notes**: Robust standard errors are within ( ). ***, **, and * denote significance levels at 1%, 5%, and 10%, respectively.

**Table 9 ijerph-19-12432-t009:** Regression results dealing with endogeneity.

Variable	Instrumental Variables Method 2SLS		Generalized Moment Estimation Method	
	Phase I dig	Phase II lnc	DIF-GMM lnc	Twostep SYS-DMM lnc
dig	2.159 ** (0.149)	2.720 *** (0.135)	2.319 *** (0.187)	2.547 *** (0.177)
iv	0.238 ** (0.0769)	-	-	-
L.lnc	-	-	0.7488 *** (0.132)	0.985 *** (0.0885)
Controls	Yes	Yes	Yes	Yes
Constant	4.991 *** (0.206)	3.838 *** (0.277)	-	0.484 (0.442)
Kleibergen–Paap rk LM	21.075 [0.000]		-	-
Kleibergen–Paap rk Wald F	9.537 {8.96}		-	-
Hansen	-	-	1	0.552
AR(1)	-	-	0.00450	0.00047
AR(2)	-	-	0.216	0.220
Obs.	240	240	210	270

**Note:***p* values are in [] and critical values in { } for the Stock–Yogo weak identification test at the 15% level.

**Table 10 ijerph-19-12432-t010:** Robustness tests of the level of digital innovation affecting carbon emissions in the transportation sector.

Variables	(1)	(2)	(3)		
	Carbon Emission Intensity	lnc	East	Central	West
Inter	-	0.862 *** (0.286)	-	-	-
dig	0.749 ** (0.4102)	-	2.500 ** (0.247)	2.809 ** (0.330)	2.482 * (0.277)
Controls	Yes	Yes	Yes	Yes	Yes
Constant	5.699 *** (0.0784)	3.479 *** (0.531)	4.386 *** (0.411)	8.046 *** (0.825)	3.359 *** (0.731)
Obs.	270	270	99	81	90
R-squared	0.028	0.121	0.465	0.495	0.479
Number of id	30	30	11	9	10

**Notes:** Robust standard errors are within ( ). ***, **, and * denote significance levels at 1%, 5%, and 10%, respectively.

**Table 11 ijerph-19-12432-t011:** Robustness test of the spatial spillover effect of digital innovation level affecting carbon emissions in the transportation sector.

Model Setting		Sar		
Spatial Matrix Type		Economic Distance	Geographical Distance	Adjacency Matrix
Variable		(1)	(2)	(3)
rho		−0.191 ** (0.0870)	−0.138 ** (0.0890)	−0.112 * (0.0607)
dig		0.427 *** (0.153)	0.392 ** (0.154)	0.433 *** (0.154)
dig2		−0.0600 ** (0.0252)	−0.0540 ** (0.0253)	−0.0599 ** (0.0253)
Controls		Yes	Yes	Yes
Direct effect	dig	0.436 *** (0.159)	0.398 ** (0.158)	0.440 *** (0.159)
	dig2	−0.0617 ** (0.0262)	−0.0554 ** (0.0261)	−0.0613 ** (0.0261)
Indirect effects	dig	0.0719 * (0.0437)	0.0386 (0.0743)	0.0455 (0.0315)
	dig2	−0.0102 (0.00667)	−0.0054 (0.0107)	−0.0064 (0.00466)
Total effect	dig	0.508 *** (0.131)	0.437 ** (0.156)	0.486 *** (0.141)
	dig2	−0.0719 ** (0.0216)	−0.0608 ** (0.0253)	−0.0677 ** (0.0232)
LogL		211.4288	209.3413	210.7885
R-squared		0.071	0.083	0.088

**Notes:** Robust standard errors are within ( ). ***, **, and * denote significance levels at 1%, 5%, and 10%, respectively.

**Table 12 ijerph-19-12432-t012:** Analysis of the mechanisms by which the level of digital innovation affects carbon emissions in the transport sector.

Variables’ Moderator	Traffic Development Level	Transportation Energy Consumption Structure
dig2 * moderator	0.0028 * (1.238)	−0.0057 ** (−2.490)
dig	0.6066 (0.539)	0.7594 (1.583)
dig2	−0.0422 * (−0.023)	−0.0640 * (−0.031)
W * dig2 * moderator	0.0045 * (1.381)	−0.0092 *** (−2.751)
W*dig	3.9163 * (1.721)	3.3268 * (1.192)
Controls	Yes	Yes
Time Effect	Yes	Yes
Spatial effects	Yes	Yes
Obs.	270	270
LogL	227	228
R-squared	0.081	0.071

**Note:** t values are in ( ). ***, **, and * denote significance levels at 1%, 5%, and 10%, respectively.

## Data Availability

Not applicable.

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
