# Peer review of "A Study on the Non-Linear Impact of Digital Technology Innovation on Carbon Emissions in the Transportation Industry"

_ijerph, 2022, doi:10.3390/ijerph191912432_

Round 1
Reviewer 1 Report
Explain the methodology better. The conclusion is good. It highlights clearly the differences in regions within China.
Author Response
Please see the attachment, Thanks!

Reviewer 2 Report
The paper has a very interesting and actual topic. The results of the paper are highly contributing to the literature. The research question is well emphasized since the beginning of the paper and the authors further follows a logical structure for dealing with it. The methodology is very well explained and presented. I appreciated the presence of the robustness tests, which further strengthen the results of the paper.
The authors should mention from the very beginning of the Abstract the sample on which the empirical analysis was carried out and the time period (the 30 provinces in China, during 2010-2018).
I also have not understand if the variable concerning digital innovation level is a composite indicator of all the variables indicated in Table 2, or if these are employed one after another in the model. It would be useful to have a brief explanation here.
The paper is well written, just some minor English language improvements are needed.
Author Response
Please see the attachment, Thanks!

Reviewer 3 Report
This manuscript employs some models to help empirically examine the impact and mechanism of digital innovation on carbon emissions in the transportation industry. The reviewer suggests that the manuscript needs a major revision before publication based on the following comments.
(1) It would be better if more quantitative analysis could be provided to the "Abstract" and "5. Conclusions and Policy Recommendations".
(2) What is the main novelty and significance of this research? It should be clearly and strongly presented in the last several paragraphs of the "1. Introduction".
(3) What progress was made against the most recent state-of-the-art similar studies, especially in the research method and model?
(4) The "5. Conclusions and Policy Recommendations" clearly present the main findings of this study. However, it would be better to make the conclusions more concise.
(5) English language should be double-checked.
Author Response
please see the attachment, Thanks!

Round 2
Reviewer 3 Report
The authors have revised the manuscript according to the reviewer's comments. Hence, the reviewer recommended that the manuscript can be published.